# Sparse Weight Activation Training

## Abstract

Training convolutional neural networks (CNNs) is time consuming. Prior work has explored how to reduce the computational demands of training by eliminating gradients with relatively small magnitude. We show that eliminating small magnitude components has limited impact on the direction of high-dimensional vectors. However, in the context of training a CNN, we find that eliminating small magnitude components of weight and activation vectors allows us to train deeper networks on more complex datasets versus eliminating small magnitude components of gradients. We propose Sparse Weight Activation Training (SWAT), an algorithm that embodies these observations. SWAT reduces computations by 50% to 80% with better accuracy at a given level of sparsity versus the Dynamic Sparse Graph algorithm. SWAT also reduces memory footprint by 23% to 37% for activations and 50% to 80% for weights.

## 1 Introduction

The usage of convolutional neural networks (CNNs) has dominated a wide variety of complex computer vision tasks, such as object recognition (Krizhevsky et al., 2012; Szegedy et al., 2015), object detection (Szegedy et al., 2013; Ren et al., 2015), and image restoration (Dong et al., 2014; Zhang et al., 2017). However, CNNs are compute and memory intensive; even a moderately sized CNN model, like ResNet-50 with tens of millions of parameters, requires billions of floating-point operations and consumes tens of gigabytes to store weights and activations during training.

Previous works propose techniques for reducing computations and memory consumption during CNN training. Such techniques include quantization where every operation is quantized in low-precision during training such a (Zhou et al., 2016; Choi et al., 2018; Wu et al., 2016; Wang et al., 2018), or, use fixed-point integers instead of floating-point numbers (Wu et al., 2018; Das et al., 2018).

An orthogonal approach to reduce computations is sparsification, a process in which we eliminate computations involving small values. meProp (Sun et al., 2017; Wei et al., 2017) sparsifies back-propagating by selecting a subset of output gradients in each layer. Using only the top 5% of the gradients (ranked by magnitude), meProp can train a CNN and MLP on MNIST dataset without accuracy loss. The computational flow of meProp is shown in Figure 1a and 1b. meProp does not modify the forward pass. In the backward pass meProp performs a "Top-K" operation on the output activation gradients which sets components not ranked in the Top-K by magnitude to zero. It then uses the sparsified output activation gradients to (potentially more efficiently) compute the input activation and weight gradients. Our experiments suggest meProp fails to converge on larger networks and datasets.

Recently, Liu et al. (2019) proposed a method of reducing computation during training and inference by constructing a dynamic sparse graph (DSG) using random projection for dimensionality reduction. DSG loses accuracy on ImageNet dataset.

In this work, we propose an alternative technique, Sparse Weight Activation Training (SWAT), that can train deep CNNs on complex data sets like ImageNet. Compared to DSG, SWAT is a straightforward technique which uses less expensive Top-K operation, inspired by meProp, while achieving better accuracy than DSG on ImageNet.

This paper provides the following contributions:

- It shows that dropping gradients during back-propagation is harmful to network convergence especially when training a deeper model on a complex dataset. In this case the model suffers high accuracy loss.

- It proposes SWAT, a sparse training algorithm that can train a broad range of deep CNNs with minimal accuracy loss on complex datasets like CIFAR10, CIFAR100, and ImageNet. SWAT reduces the total number of operations during training by 50%–80%. It also achieves 23%–37% activation and 50%–80% weight footprint reduction during the backward pass.

- SWAT algorithm uses sparse weight both in the forward and backward passes, and therefore model learns sparse weights, i.e., a pruned architecture; If the model has been trained using SWAT with $S\%$ sparsity during training, then during inference, weight can be pruned to $S\%$ without sacrificing any loss in accuracy.

- We perform empirical studies to provide insight into why 'SWAT performs well; we showed that Top-K sparsification in general preserves direction in high-dimensional space.

## 2 SPARSITY INDUCED TRAINING

### 2.1 PRELIMINARIES

Let us consider a deep CNN with L convolutional layers trained using mini-batch stochastic gradient descent, where the $l^{th}$ layer maps the input activation $(a_{l-1})$ using function $f_l$ from $R^{N \times C_{l-1} \times H_{l-1} \times W_{l-1}} \rightarrow R^{N \times C_l \times H_l \times W_l}$. $f_l$ computes $C_l$ channel of output feature maps, each of dimension $R^{H_l \times W_l}$, using $C_{l-1}$ channels of input feature maps of dimension $R^{H_{l-1} \times W_{l-1}}$ for each of the $N$ samples in the mini-batch. The $l^{th}$ layer has weights $w_l \in R^{C_l \times C_{l-1} \times H_f \times W_f}$. The forward pass of the $l^{th}$ layer can be defined as:

$$a_l = f_l(a_{l-1}, w_l) \tag{1}$$

During back-propagation the $l^{th}$ layer receives the gradient of the loss $L$ w.r.t its output activation $(\nabla_{a_l})$. This is used to compute the gradient of the loss w.r.t its input activation $(\nabla_{a_{l-1}})$ and weight $(\nabla_{w_l})$. Thus, the backward pass for the $l^{th}$ layer can be defined as:

$$\nabla_{a_{l-1}} = F_l(\nabla_{a_l}, w_l) \tag{2}$$

$$\nabla_{w_l} = F_l(\nabla_{a_l}, a_{l-1}) \tag{3}$$

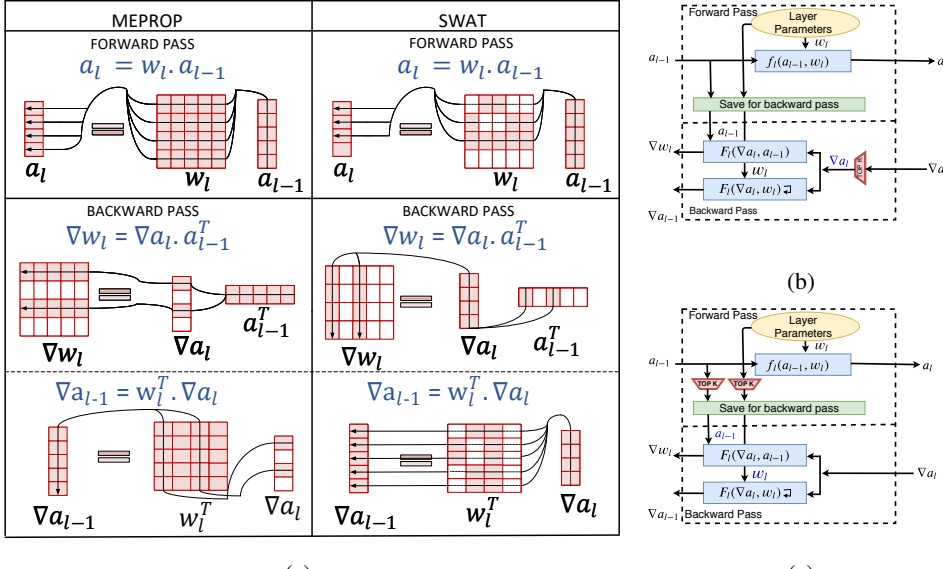

(a)  (c)

Figure 1: **meProp versus SWAT** (a) Shows the forward and backward pass of MeProp and SWAT for a fully connected layer. (b) Computational flow of meProp for any layer $l$ (c) Computational flow of SWAT for any layer $l$.

## 2.2 SPARSE WEIGHT ACTIVATION TRAINING

Our goal is to reduce the computations required during the training process. SWAT does this by effectively transforming small magnitude components of vectors into zero values. Since multiplication of any number by zero results in zero the multiplication is not necessary. Such a sparsification process can be applied in a number of ways in the context of the backpropagation algorithm. Ideally, the modified training algorithm will retain both model accuracy and rate of convergence.

We explore the sensitivity to applying this sparsification process at different points. We look at the sensitivity of the model convergence to sparse input i.e. weight ($w_l$) and input activation ($a_l$) for forward pass and weights ($w_l$), input activation ($a_l$) and output activation gradient ($\bigtriangledown_{a_l}$) for the backward pass as shown in Equation 1 , 2 and 3.

Figure 2a shows the result of our analysis of sparsification in the forward pass. Here we train ResNet-18 on CIFAR-100 after modifying the forward pass to use sparse inputs in Equation 1 for weights ($w_l$) and separately activations ($a_l$) while keeping the backward pass unchanged (i.e., only the forward pass computation is sparse). The results show that network convergence is more tolerant to sparse weights ($w_l$) compared to sparse activations ($a_l$). Thus, as shown in Figure 1a and 1c, SWAT uses sparse weights in the forward pass.

Figure 2b shows the result of our analysis of sparsification in the backward pass. We modify Equation 2 and Equation 3 to sparsify either output gradient ($\bigtriangledown_{a_l}$), as in meProp (see Figure 1b), or sparsify activations ($a_l$) and weights ($w_l$). The results show accuracy is extremely sensitive to sparsification of output gradients. Such sparsity consistently results in networks converging to lower accuracy compared to using sparse activations and weights. Thus, as shown in Figure 1a and 1c, SWAT uses sparse weights and activations in the backward pass. The overall SWAT training algorithm is presented in Algorithm 1.

SWAT uses sparse computation in both the forward and the backward passes, while meProp (Sun et al., 2017) uses sparse computation only in the backward pass. SWAT uses sparse weights and activations in the backward pass allowing compression of weights and activations in the forward pass[1]. Effectively, reducing overall memory access overhead of fetching weights in the backward pass and activation storage overhead because only Top-K% are saved. This memory benefit is not present for meProp since dense weights and activations are needed in the backward pass, whereas there is no storage benefit of sparsifying the output gradients since they are temporary values generated during back-propagation.

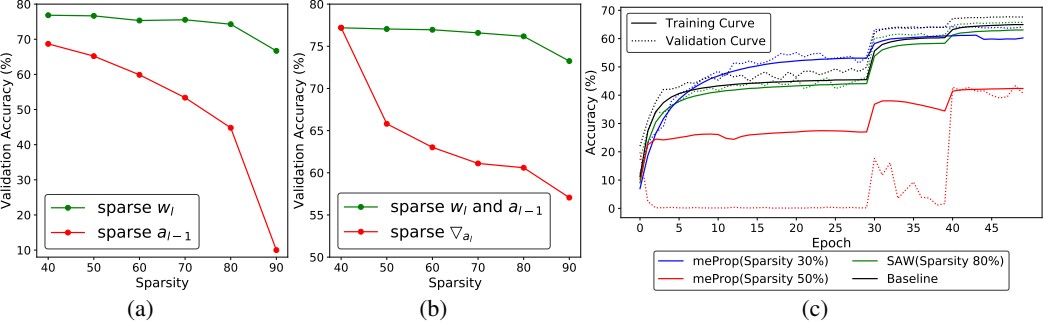

Figure 2: **Convergence Analysis**: (a) Sensitivity Analysis of ResNet18 for the Forward Pass on the CIFAR100 dataset. (b) Sensitivity Analysis of ResNet18 for the Backward Pass on the CIFAR100 dataset. (c) Shows the training curve of ResNet18 on ImageNet for meProp and SAW algorithm. Learning rate is reduced by $\frac{1}{10}^{th}$ at $30^{th}$ and $40^{th}$ epoch.

To compare SWAT's approach to that of meProp, we use a variant of SWAT that only sparsifies the backward pass; we shall refer to this version of SWAT as SAW (Sparse Activation and Weight back-propagation). We compare the performance of the meProp and SAW with deep networks, and complex datasets[2]. Figure 2c shows SAW and meProp convergence of ResNet18 with the ImageNet dataset; it compares the performance of meProp at 30% and 50% sparsity to SAW 80% sparsity. As we can see, meProp converges to a good solution at sparsity of 30%. However, at 50% sparsity,

---

[1]more detail in section 3.3

[2]The performance of SWAT with deep networks and complex datasets is in the result section.

meProp suffers from overfitting and fails to generalize (between epochs 5 to 30), and at the same time, it is unable to reach an accuracy level above 45%. These results suggest that dropping output activation gradient ($\nabla_{a_l}$) is generally harmful during back-propagation. On the other hand, SAW succeeds to converge to an accuracy of 64% even at a much higher sparsity of 80%.

---

**Algorithm 1:** Training an $L$ layer network using SWAT or SAW Algorithm

---

**The data:** A mini-batch of inputs & targets ($\mathbf{a}_0, \mathbf{a}^*$), training iteration $t$, previous weights $\mathbf{w}^t$, learning rate $\eta$.

**The result:** Update weights $\mathbf{w}^{t+1}$.

Step 1. Forward Computation;

**for** $l = 1\ to\ L$ **do**
    **if** $l == $ '*ConvolutionLayer*' *or* '*LinearLayer*' **then**
        **if** *algorithm* == '*SWAT*' **then**
            $\mathbf{w}_l^t \Leftarrow f_{TOPK}(w_l^t)$;
            $\mathbf{a}_l^t \Leftarrow \texttt{forward}(\mathbf{w}_l^t, \mathbf{a}_{l-1}^t)$;
            $\mathbf{a}_{l-1}^t \Leftarrow f_{TOPK}(a_{l-1}^t)$;
        **else**
            // algorithm == 'SAW';
            $\mathbf{a}_l^t \Leftarrow \texttt{forward}(\mathbf{w}_l^t, \mathbf{a}_{l-1}^t)$
            $\mathbf{w}_l^t \Leftarrow f_{TOPK}(w_l^t)$;
            $\mathbf{a}_{l-1}^t \Leftarrow f_{TOPK}(a_{l-1}^t)$;
        **end**
        **save_for_backward**$_l \Leftarrow \mathbf{w}_l^t, \mathbf{a}_{l-1}$;
    **else**
        $\mathbf{a}_l \Leftarrow \texttt{forward}(\mathbf{w}_l^t, \mathbf{a}_{l-1})$;
        **save_for_backward**$_l \Leftarrow \mathbf{w}_l^t, \mathbf{a}_{l-1}$;
    **end**
**end**

Step 2. Backward Computation;
Compute the gradient of the output layer
$$\nabla_{\mathbf{a}_L} = \frac{\partial loss(\mathbf{a}_L, \mathbf{a}^*)}{\partial \mathbf{a}_L};$$
**for** $l=L\ to\ 1$ **do**
    $\mathbf{w}_l^t, \mathbf{a}_{l-1} \Leftarrow$ **save_for_backward**$_l$;
    $\nabla_{\mathbf{a}_{l-1}} \Leftarrow \texttt{backward\_input}(\nabla_{\mathbf{a}_l}, w_l^t)$;
    $\nabla_{\mathbf{w}_{l-1}} \Leftarrow$
      $\texttt{backward\_weight}(\nabla_{\mathbf{a}_l}, a_{l-1}^t)$;
**end**

Step 3. Parameter Update;
**for** $l=1\ to\ L$ **do**
    $\mathbf{w}_l^{t+1} \Leftarrow Optimizer(\mathbf{w}_l^t, \nabla_{\mathbf{w}_l}, \eta)$;
**end**

---

**Top-K Selection:** Given CNNs operate on tensors with many dimensions, there are several options for how to select which components are set to zero during sparsification. Our CNNs operate on fourth-order tensors, $T \in R^{N \times C \times H \times W}$. Below we evaluate three variants of the Top-K operation illustrated in the right side of Figure 3. We also compared against a null hypothesis in which randomly selected components of a tensor are set to zero.

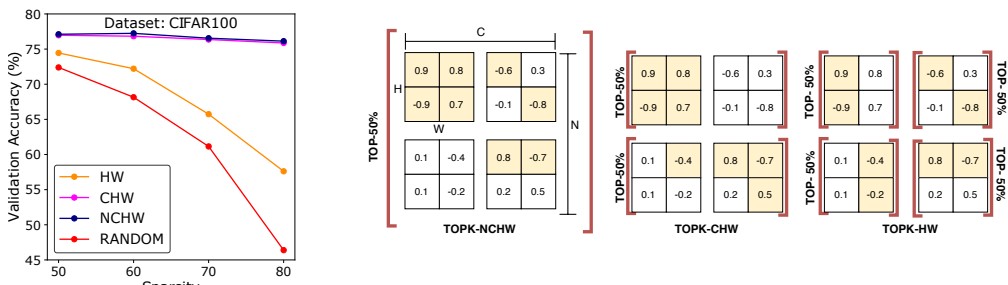

Figure 3: **Different ways of performing top-k operation**. 'N' denotes the #samples in the mini-batch or filters in the layer, 'C' denotes the #channels in the layer. 'H' and 'W' denote the height and width of the filter/activation map in the layer. Color represent the selected activations/weights by the Top-K operation.

The first variant, labeled TOPK-NCHW in Figure 3, selects activations and weights to set to zero by considering the entire mini-batch. This variant performs Top-K operation over the entire tensor, $f_{TOPK}^{\{N,C,H,W\}}(T)$, where the superscript represents the dimension along which the Top-K operation is performed. The second variant (TOPK-CHW) performs Top-K operation over the dimensions $C, H$ and $W$ i.e., $f_{TOPK}^{\{C,H,W\}}(T)$, i.e., selects K % of input activations from every mini-batch sample and K% of weights from every filter in the layer. The third variant (TOPK-HW) is the strictest form of

Top-K operation. It select K% of activations or weights from all channels, and thereby performing the Top-K operation over the dimension $H$ and $W$, i.e., $f_{TOPK}^{\{H,W\}}(T_{H,W})$.

The left side of Figure 3 shows the accuracy achieved on ResNet-18 for CIFAR100 when using SAW configured with each of these Top-K variants along with a variant where a random subset of components is set to zero. The results show, first, that randomly selecting works only for low sparsity. At high sparsity all variants of Top-K outperform random selection by a considerable margin. Second, they show that the more constrainted the Top-K operation the less accuracy achieved. Constraining Top-K results in selecting some activations or weights which are quite small. Similarly, some essential activations and weights are discarded just to satisfy the constraint.

## 3   RESULTS

In this section, we present our experimental results of SWAT algorithm on different architectures and datasets and we quantify the theoretical reduction in compute and memory bandwidth achievable using SWAT.

### 3.1   EXPERIMENTAL SETUP

We implement SWAT and SAW algorithms in PyTorch Framework (Paszke et al., 2017); models are trained on three different datasets: CIFAR10, CIFAR100 (Krizhevsky et al., 2009) and ImageNet ILSVRC2012 (Deng et al., 2009) and are evaluated on four different architectures ResNet-18, 34, 50, 101 (He et al., 2016), Wide Residual Networks (Zagoruyko & Komodakis, 2016), DenseNet-BC-121 (Huang et al., 2017), and VGG-16 (Simonyan & Zisserman, 2014) with batch-normalization (Ioffe & Szegedy, 2015). Batch-Normalization statistics are computed using the running average (with momentum 0.9). We use SGD with momentum as our optimization algorithm with an initial learning rate of 0.1, momentum of 0.9 and weight decay $\lambda$ of 0.0001.

For CIFAR10 and CIFAR100 dataset, ResNet, VGG, and DenseNet models are trained for 150 epochs, and learning rate are reduced by $(1/10)^{th}$ at the 50-th and the 100-th epoch whereas WRN is trained for 200 epochs and the learning rate is annealed by a factor of $(1/5)^{th}$ at 60-th, 120-th and 160-th epoch. ResNet, VGG, and WRN are trained using a batch-size of 128 whereas DenseNet is trained with a batch-size of 64. We run each experiment with three different seeds and use the average value for all the plots.

For training on ImageNet dataset, we use $224 \times 224$ random crops from the input images or its horizontal flip and the input image is normalized by the per-color mean and standard deviation. Networks are trained for 50 epochs with the mini batch-size of 256 samples, and the learning rate are reduced by $(1/10)^{th}$ after 30-th and 40-th epoch.

### 3.2   ACCURACY ANALYSIS

In this section, we provide a comprehensive analysis of SWAT and SAW algorithms and show the influence of sparsity on validation accuracy; thereby showing the potential of reducing computation during training with negligible accuracy loss. We furthermore discuss the impact on rate of convergence and the robustness of the algorithm on a wide range of models with different depths and widths. Last, we provide an alternative to Top-K for efficient software/hardware implementation.

**Accuracy on CIFAR10 and CIFAR100:** Figure 4 shows the accuracy of the SWAT and SAW algorithms at different sparsity budgets on CIFAR10 and CIFAR100 dataset. From the graph, we can conclude that models can be trained using SWAT and SAW algorithm up to 60% sparsity with almost zero accuracy loss and suffer only a slight accuracy loss at 70% sparsity. For CIFAR10 dataset at 70% sparsity, VGG-16 and DenseNet-121 have an accuracy loss of around 0.57% for SWAT (0.26% for SAW) and 0.4% for SWAT (0.23% for SAW) whereas ResNet-18 gains an accuracy of 0.02% for SWAT (0.1% for SAW). For CIFAR100 dataset at 70% sparsity, ResNet-18, VGG-16 and DenseNet-BC-121 lose an accuracy of around 0.5%, 0.41% and 0.68% for SAW and 0.4%, 0.99% and 1.78% for SWAT respectively. At 80% sparsity the accuracy loss on CIFAR10 and 100 is less than 1.8% for SAW and less than 2.5% for SWAT.

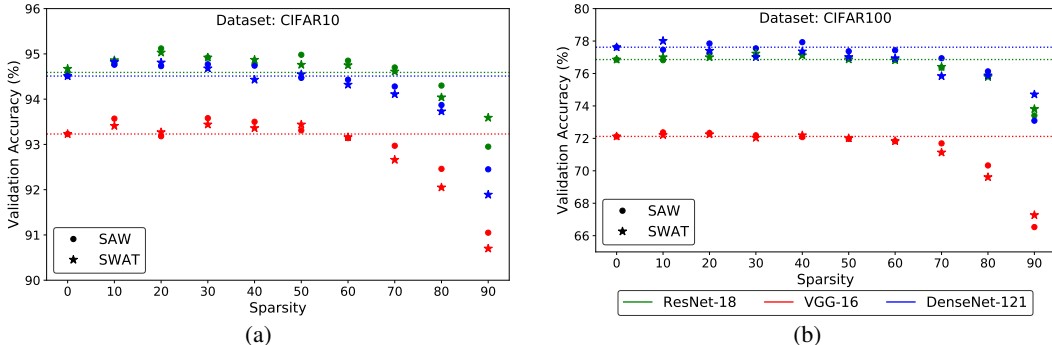

(a)  (b)

Figure 4: **Comprehensive analysis of sparsity vs accuracy trade-off**: (a) Accuracy of SWAT and SAW algorithms on CIFAR10 dataset. (b) Accuracy of SWAT and SAW algorithms on CIFAR100 dataset. The dashed line represents the baseline accuracy for the corresponding model. Datapoints for SAW algorithm are represented as dots whereas for SWAT algorithm they are represented as stars.

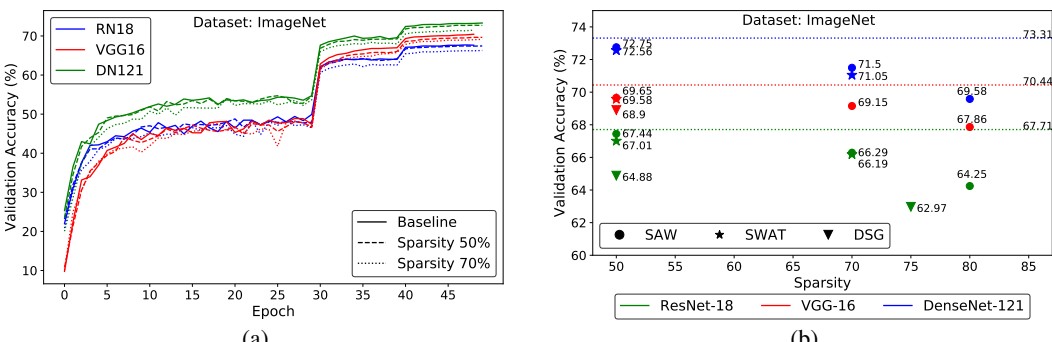

(a)  (b)

Figure 5: **Trend of SWAT algorithm on ImageNet dataset**: (a) Validation curve of SWAT algorithm (b) Validation Accuracy of SWAT, SAW and DSG algorithms at different sparsity constraints. Dotted line represents the baseline back-propagation algorithm. 'RN18' represent ResNet-18, 'DN121' represent DenseNet-BC-121 and 'DSG' denote the results reported by the Dynamic Sparse Graph(Liu et al., 2019) algorithm.

**Accuracy on ImageNet:** Figure 5a shows the validation curve of the SWAT algorithm on ImageNet dataset for three different architectures and Figure 5b shows the accuracy obtained by the SWAT and SAW algorithms. The result shows that the SWAT and SAW algorithms lose negligible accuracy at $50\%$ sparsity for all three architectures. The solution is within an accuracy loss of $0.26 - 1.01\%$ compared to the baseline solution. For high sparsity of $70\%$, ResNet-18, VGG-16 and DenseNet-BC-121 lose only around $1.52\%$, $1.6\%$ and $2.26\%$ accuracy for the SWAT and $1.42\%$, $1.28\%$ and $1.82\%$ for the SAW algorithm respectively. Both the algorithms perform better than the DSG algorithm proposed by (Liu et al., 2019), which accelerates training by performing dimensionality reduction search and performing the forward and backward passes in low dimensional space. The accuracy loss of DSG at $50\%$ sparsity is around $2.83\%$ for ResNet-18 and $1.54\%$ for VGG-16 compared to the SWAT accuracy loss of $0.27\%$ and $0.86\%$ for ResNet-18 and VGG-16 respectively.

**Impact on rate of Convergence:** We define the rate of convergence is the number of epochs it takes to reach the saturation accuracy. Figure 5a shows the validation curve of SWAT algorithm when training ResNet-18, VGG-16 and DenseNet-BC-121 on ImageNet dataset. As shown in Figure, when the learning rate is 0.1 (i.e. between epoch 0 and 30) the SWAT algorithm reaches the saturation accuracy around the 15th epoch approximately the same epoch when the baseline algorithm also reaches saturation. Similarly, when the learning rate is 0.01 (i.e. between epoch 0-40th) both SWAT and the baseline saturate at epoch 35th. The experiment at $50\%$ and $70\%$ sparsity shows that SWAT algorithm converges with slight accuracy loss but at the same rate compared to the baseline algorithm.

**Influence of Depth and Width:** Network depth ($\#layers$) and width ($\#channels$) are two important design criteria. Previous studies (Lu et al., 2017; Raghu et al., 2017; Sharir & Shashua, 2018) have found that both depth and width affect network expressivity. Increasing network depth helps in

learning complex abstraction, whereas increasing width helps in learning more features. Ideally, we want SWAT and SAW algorithms to work with models of varying depth and width. Therefore, we study the influence of depth and width on the SWAT and SAW algorithms. Figure 6a shows the accuracy of ResNet-50, ResNet-101 and WRN-28-10 on CIFAR100 datasets at four different sparsities $0\%$, $50\%$, $70\%$ and $80\%$. The result for deeper networks shows that enforcing sparsity up to $70\%$ is beneficial for training as the ResNet-50 and ResNet-100 converged to an accuracy higher than the baseline training. At $80\%$ sparsity, ResNet-101 loses accuracy of a mere $0.18\%$ whereas ResNet-50 still has an accuracy advantage of $0.19\%$ over the baseline. WRN-28-10 lose accuracy on training with SWAT and SAW algorithm, but the accuracy loss is only $0.67\%$ for SAW and $0.49\%$ for SWAT at $70\%$ sparsity.

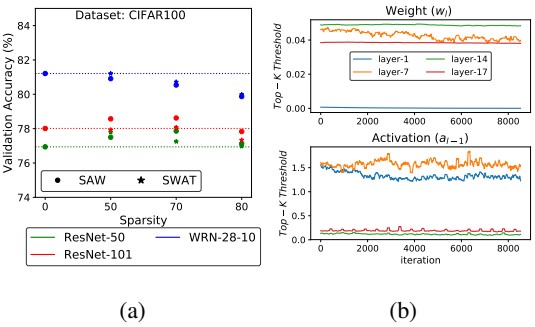

(a)                                     (b)

Figure 6: (a) **Influence of Depth and Width**: Accuracy of SAW and SWAT algorithms on CIFAR100 dataset for ResNet-50,101 and WRN-28-10 **(b) Threshold value (K-th largest values) in Top-K operation of different layers during training**

**Efficient Top-K Implementation:** Top-K[3] operation on 1 dimensional array of size $n$ can be naively implemented using sorting. The computational complexity of a naive Top-K operation is $O(n \log n)$. The computational complexity can be reduced to $O(n)$, if the k-th largest element can be found in $O(n)$ time, since for this case the Top-K operation can be implemented by a threshold operation. The K-th largest element can be computed in $O(n)$ average time using quickselect (Hoare, 1961) or in $\theta(n)$ time using BFPRT (Blum et al., 1973) or introselect (Musser, 1997). The computation can be further reduced since we found experimentally that for a given layer, the K-th largest elements is almost constant during training as shown in Figure 6b. So we don't need to compute the K-th largest elements during every training iteration and can be computed once in a while after multiple iterations.

### 3.3 COMPUTATIONAL AND MEMORY OVERHEAD REDUCTION DURING TRAINING

In this section, we will quantify the reduction in computational and memory overhead, using SWAT, over the baseline training algorithm.

**Computation Reduction:** SWAT algorithm is accelerating CNN training by sparsifying the computation in both the forward and the backward pass. During CNN training, most of the computation is in the convolution and fully connected layer, therefore sparsifying the computation in both of these layers can result in linear speed-up during training. Figure 7 shows the computational reduction possible by SWAT for three different architecture while training on ImageNet dataset. SWAT achieves a computation reduction of 2x, 3.3x, and 5x at 50%, 70%, and 80% sparsity respectively. Note that the overall overhead of implementing efficient Top-K operation using BFRT/introselect + thresholding, as described in the previous section, is only 1-2% additional computation during training. Another benefit of using SWAT

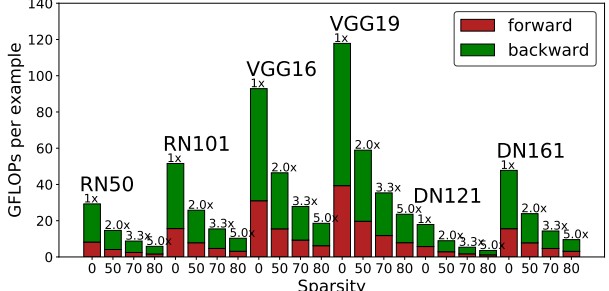

Figure 7: **Computational reduction in SWAT at different sparsity**. "RN" denotes ResNet, "DN" denote DenseNet (Dataset: ImageNet)

---

[3]Top-K operation is performed on an absolute values.

is that the model learns a sparse architecture and therefore, sparse weights are used during Inference. Thus, the same computational benefit of 2-5x is possible for Inference as well.

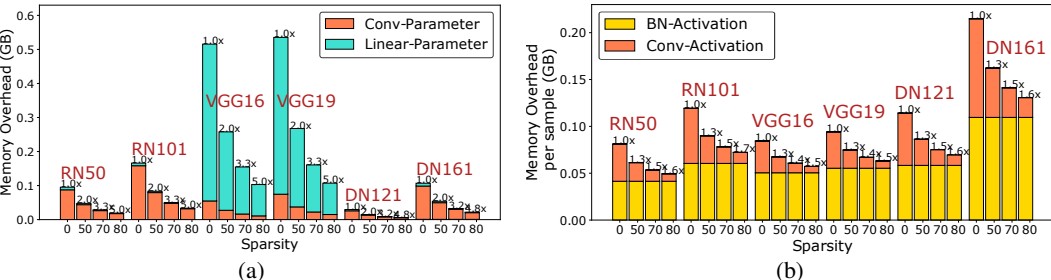

Figure 8: **Reduction in memory accesses during the backward pass**. (a) Reduction in parameter access (b) Reduction in activation access per sample (Dataset: ImageNet)

**Memory Overhead Reduction:** During training, most of the weights and activations are stored in DRAM and accessing DRAM consumes three orders of magnitude more energy consumption than computation (Horowitz). So reducing the memory access during training will directly reduce the energy consumption. SWAT algorithm uses sparse input activation ($a_{l-1}$) and weight ($w_{l-1}$) in the backward, so input activation and weight can be compressed and stored in the memory in sparse format thereby reducing the DRAM access in the backward pass. Figure 8a shows the reduction of 2x, 3.3x, and 5x at sparsity of 50%, 70% and 80% in the parameters access in the backward pass. The other significant memory overhead is saving the activation, and this overhead is dependent not only on the model size but also on the batch-size used during training. Figure 8b shows the activation memory overhead for different architectures for a mini-batch size of 1. The graph only shows the activation of batch-normalization and convolutional layer since memory overhead for the activations of the linear layer is negligible compared to them. Note that SWAT algorithm sparsifies the computation of the linear, and convolutional layers, so full input activation of the batch-normalization layer are saved for the backward pass. Thus, SWAT algorithm achieves activation compression of around 1.3x, 1.5x, and 1.7x at sparsity of around 50%, 70%, and 80%. The activation of batch-normalization layer limits the overall activation compression.

## 4 EXPERIMENTAL ANALYSIS OF SWAT BEHAVIOUR

In this section, we will give some experimental evidence which explains why the SWAT algorithm is working so well in practice. The experiment shows the general behavior of vector sparsification in high dimensional space. Let us first define two new terminologies which we are going to use in this section: "Top-K sparsification" and "Sparsification Angle"[4]. Top-K sparsification of a vector $v$ selects K% of the highest magnitude component and set the rest of the component to zero. Sparsification angle is the angle between the original and the Top-K sparsified instance of that vector.

### 4.1 VECTOR SPARSIFICATION IN HIGH-DIMENSIONAL SPACE

A vector in high-dimensional space behaves differently from their low dimensional counterpart. It is well known that in high-dimension, two independent isotropic vectors tend to be orthogonal. Recently Anderson & Berg (2017) extended the analysis of the geometry of high-dimensional vectors to binary vectors. They proved that the angle between any random vector, drawn from a rotationally invariant distribution, and its binarized version would be concentrated around $37°$. Thus, binarization of high dimensional vector approximately preserves their direction. We apply a similar kind of geometry analysis to high dimensional sparsified vectors. We show that sparsification indeed preserves direction in high-dimensional space, which is contrary to our low-dimensional intuition.

The first question we need to answer is how should we sparsify a vector $v \in R^d$ such that the sparsification angle is minimum between the original and the sparsified instance of the original vector (has only $K\%$ non-zero component). We found that the cosine angle is minimum when the $K\%$ non-zero component corresponds to the $K\%$ highest magnitude component, i.e., Top K sparsification. The proof is in the appendix.

---

[4]The mathematical definition of both of these terms is in the appendix A

We did an experiment for analyzing how Top K sparsification affect angle as shown in Figure 9a, Here we are showing the sparsification angle distribution for a 1000 dimensional vector drawn from standard normal distribution at different sparsity. The peak of sparsification angle at 90% sparsity is concentrated around 48° which is much less than peak of random vectors which is concentrated around 90°. Similarly, the peak up to 80% sparsity is concentrated at an angle of 36.4° only. This suggest that deviation caused by sparsification is indeed small in high-dimension.

For our next experiment shown in Figure 9b, we study how much a vector of a given dimension, drawn from standard normal distribution, can be maximally sparsified such that the sparsification angle is less than $\delta$. We can see that the percentage of Top-K components needed for $\delta = \{20°, 30°, 40°\}$ is around only 43%, 29% and 18% respectively with a variance less than 3% as shown in Figure 9c. Thus, these experiment suggest that a high-dimensional vector can be sparsified up to 70%, which will lead a deviation ($\delta$) of only 30°. All the above experimental results are dependent on the distribution from which random vectors are drawn, in Figure 9e, we calculate the sparsification angle during training ResNet18 on CIFAR100 dataset at 70% Sparsity. Here the sparsification angle for weight and activation is less than 36° for all the layer in the network. So the experiment suggests that the above analysis is applicable during training.

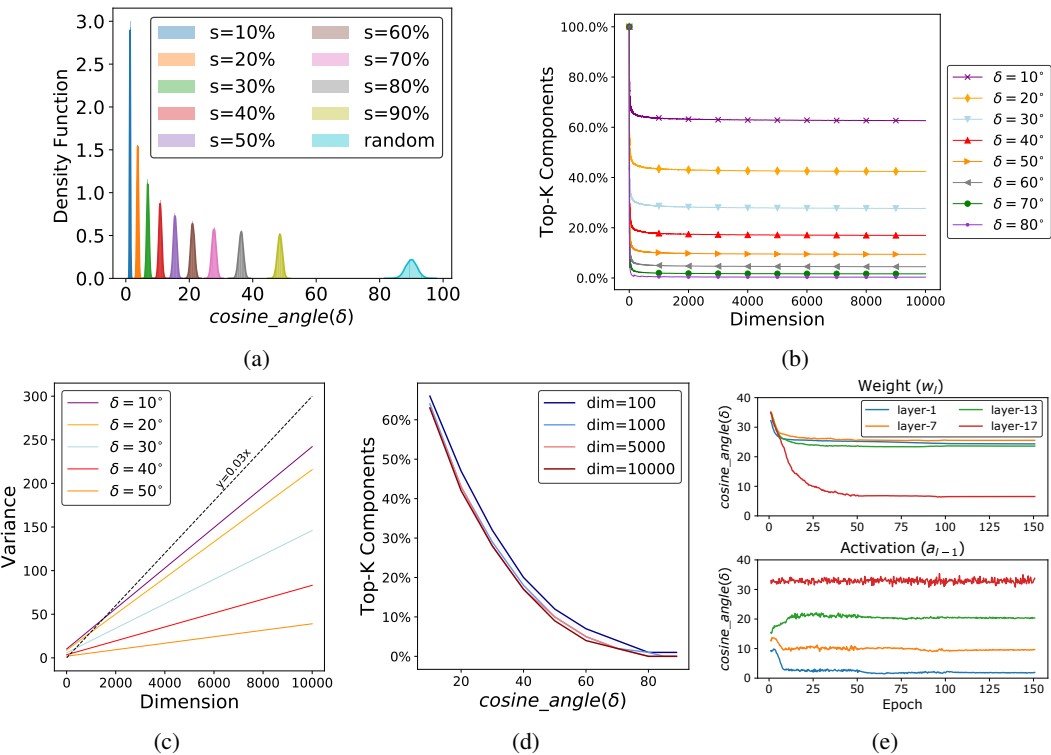

Figure 9: **Vector sparsification in high-dimension approximately preserves the direction**:(a) Shows the sparsification angle distribution at different Top-K percentage for a 1000 dimensional vector in 10, 000 trials. Random represents the angle distribution between 2 random vectors. (b) and (c) Shows the percentage of Top-K components needed for sparsification angle to be within $\delta$ in 1000 trials and the variance in those trials. (d) Shows the relation between Top-K sparsification and the sparsification angle in 1000 trials. (e) Shows how the sparsification angle (at 70% sparsification) varies during training for ResNet-18 architecture on CIFAR100 dataset.

## 5 RELATED WORK

We can classify most of the previous studies which focus on accelerating training or inference in the following broad categories:

**Pruning**: Most of the pruning work focuses on inference optimization. Weight pruning can be classified into two broad categories of structured and unstructured pruning. The idea of unstructured

pruning can be traced back to LeCun et al. (1990); Hassibi & Stork (1993), which prune the network using the saliency of parameters derived from the second-order information of loss function. Han et al. (2015b;a) pruned network parameters using a magnitude based method. There are several other unstructured pruning methods such as Molchanov et al. (2017); Louizos et al. (2017), but the drawback of all these methods is that it is difficult to extract parallelism on hardware. In contrast, structured pruning such as Liu et al. (2017); Li et al. (2016b); He et al. (2017); Luo et al. (2017); Wen et al. (2016); Molchanov et al. (2016) removes entire channels or filters at a time which preserves the inherent regular computation structure, and therefore it is easy to extract parallelism on hardware.

**Quantization**: Quantized networks can be used to accelerate both training and inference since energy consumption in the hardware is directly proportional to bit-width of the operands. There are many works which focus on quantizing weights for efficient inference such as McDonnell (2018); Wu et al. (2016); Zhu et al. (2016); Li et al. (2016a); Courbariaux et al. (2015) whereas much other work focuses on accelerating training as well, such as Banner et al. (2018); Choi et al. (2018); Wang et al. (2018); Lin et al. (2017a); Zhou et al. (2016); Courbariaux et al. (2016); Rastegari et al. (2016); Gupta et al. (2015). Some of the other work such as Zhao et al. (2019); McKinstry et al. (2018); Zhou et al. (2017); Mellempudi et al. (2017) shows that training from scratch is not necessary for finding the quantized model, but one can find a quantized model from pre-trained full precision models. Other work focuses on discrete training and inference using Integers such as Wu et al. (2018); Das et al. (2018); Jacob et al. (2018); Lin et al. (2016) since integer added/multiplier is more efficient than floating-point adder/multiplier. Few studies such as Louizos et al. (2019); Jung et al. (2019); Zhang et al. (2018); Zhou et al. (2018); Hou & Kwok (2018); Hou et al. (2016) formulate the quantization as an optimization problem to minimize the accuracy loss due to quantization. Few other work such as Yang et al. (2019); De Sa et al. (2018) focus on improving the learning algorithm by proposing novel stochastic averaging of the low precision iterates or using SVRG to reduce the variance and by dynamically adjusting the precision representation using bit centering. Few works instead of quantizing the entire model to a fixed bit-width focus on per tensor or parameter quantization such as Sakr & Shanbhag (2019); Khoram & Li (2018). Compared to all these works, our work is orthogonal as we eliminate the computation instead of reducing the computation precision.

**Tensor Decomposition and Dimentionality Reduction**: There are few works on compressing the models by performing tensor decomposition or by learning compact structure. Alvarez & Salzmann (2017) introduce a regularizer that promotes the parameter matrix to have a low rank. Thus the algorithm encourages the model to learn a compact structure by accounting for compression during the training itself. Novikov et al. (2015) showed that tensor decomposition could be used to compress a fully connected layer by using only a few parameters. Later, Garipov et al. (2016) extended it for the convolutional layer. The idea was to reshape the kernel into a tensor of higher-order and then to factorize it.

**Distributed Training**: There are few works (Stich et al., 2018; Lin et al., 2017b) which look at reducing the communication overhead in distributed training by transferring only sparse gradients during gradient aggregation step but these works are accumulating the rest of the gradients locally for subsequent iterations. Compared to all these works, our work objective is different as we are concerned with accelerating single node training, whereas their objective is minimizing communication during distributed training.

## 6 CONCLUSION

In this work, we propose SWAT, a robust training algorithm based on the insight that sparsifying weights and activation during training has little impact on convergence. SWAT sparsify both the forward and the backward passes, thereby eliminating lots of redundant computation such as addition and multiplication by zero. SWAT is a simpler technique and performs better than the recently proposed dimensionality reduction (DSG) technique for accelerating training. Our experiments over various benchmarks demonstrate significant computation reduction of up to 2-5x for training and inference and provides a memory footprint reduction of activation by 1.3-1.7x and reduction in memory access overhead for weight by 2-5x in the backward pass.

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

## APPENDIX A    PROOF SHOWING TOP-K IS THE BEST SPARSIFICATION FUNCTION

**Definition 1** Sparsifying Function ($f_S$): Given a parameter $1 < k \leq d$, let $f_S \colon \mathbb{R}^d \to \mathbb{R}^d$ be a function that selects k component of vector and sets the rest of the component to zero. For a vector $v \in \mathbb{R}^d$, $f_S(v) = \mathbb{I}_k(\mathbf{v}) \odot \mathbf{v}$, where $\mathbb{I}_k(\mathbf{v})$ is an indicator vector having k non-zero values determined by input vector $v$.

**Definition 2** Top-K Sparsifying Function ($f_{TOPK}$): Given a parameter $1 < k \leq d$, let $f_{TOPK} \colon \mathbb{R}^d \to \mathbb{R}^d$ is a special sparsifying function that sets all but the k highest component of input vector in absolute value to zero. More precisely, for a vector $v \in \mathbb{R}^d$, $f_{TOPK}(v) = \mathbb{I}_{topk}(\mathbf{v}) \odot \mathbf{v}$ where $\mathbb{I}_{topk}$ is an indicator function, $\mathbb{I}(i \in \{\pi_1, \cdots, \pi_k\})$, and $\pi$ is a permutation of $[d]$ such that $|v|_{\pi_i} \geq |v|_{\pi_{i+1}}$ for $i = 1, \ldots, d-1$.

**Definition 3** Sparsification Angle ($\theta$): For a vector $v \in \mathbb{R}^d$, the deviation in the direction caused by sparsification $f_S(.)$ is defined as the sparsification angle, i.e., it is the angle between the vector $v$ and sparse vector $f_S(v)$.

**Lemma A.1.** *For any vector $v \in \mathbb{R}^d$ and of all the sparsifying function $f_S$, Top-K sparsifying function ($f_{TOPK}$) causes the minimum deviation in direction i.e. minimum sparsification angle.*

*Proof.* Given a parameter $k \in [1, d]$, for a vector $\mathbf{v} = (v_1, \cdots, v_n)^{\mathrm{T}} \in \mathbb{R}^d$ let $\mathbf{f_S}(\mathbf{v}) = (m_1 v_1, \cdots, m_d v_d)^{\mathrm{T}} \in \mathbb{R}^d$ such that $m_i \in \{0, 1\} \; \forall i$, be the Top-K indicator mask i.e., $m_i = 1$ only if $i^{th}$ component of $v$ is selected by the sparsifying function.

$$\cos\langle f_S(\mathbf{v}), \mathbf{v}\rangle = \frac{f_S(\mathbf{v}) \cdot \mathbf{v}}{\|f_S(\mathbf{v})\|\|\mathbf{v}\|} = \frac{\sum\limits_{i=1}^{d}(m_i v_i^2)}{\sqrt{\sum\limits_{i=1}^{d}(m_i v_i)^2}\sqrt{\sum\limits_{i=1}^{d} v_i^2}} = \frac{\sum\limits_{i=1}^{d}(m_i v_i)^2}{\sqrt{\sum\limits_{i=1}^{d}(m_i v_i)^2}\sqrt{\sum\limits_{i=1}^{d} v_i^2}} \tag{4}$$

$$= \frac{\sqrt{\sum\limits_{i=1}^{d}(m_i v_i)^2}}{\sqrt{\sum\limits_{i=1}^{d} v_i^2}} = \frac{\|f_S(\mathbf{v})\|}{\|\mathbf{v}\|} \tag{5}$$

In other words,

$$\text{Sparsifying Angle}(\theta) = \arccos\frac{\|f_S(\mathbf{v})\|}{\|\mathbf{v}\|} \tag{6}$$

arccos is a strictly decreasing function, so to minimize $\theta$, $\|f_S(\mathbf{v})\|$ must be maximized. Therefore Top-K component of the vector $v$ magnitude wise should be selected. $\qquad\square$

## APPENDIX B    PERIODIC TOP-K & EFFECT OF BATCH-NORMALIZATION

**Periodic Top-K For Efficient Implementation:**   In section 3.2, we have shown there is a little variation in the 'K-th' largest element during training, and it remains approximately constant as training proceed. Therefore, the Top-K does not need to be computed every iteration and can be periodically computed after some iterations. We define the number of iterations between computing the threshold for Top-K as the "Top-K period.

| Top-K Period | 70% Sparsity | 90% Sparsity |
|---|---|---|
| Default Top-K | 76.41 | 73.81 |
| 10 Iteration | 76.59 | 73.64 |
| 20 Iteration | 76.03 | 73.45 |
| 50 Iteration | 76.06 | 74.09 |
| 100 Iteration | 76.52 | 73.29 |

We have empirically confirmed that the periodic Top-K performs equally well as the default Top-K implementation. The table above shows top1 validation accuracy from single runs of ResNet-18 on CIFAR 100 with different Top-K periods (i.e., Top-K is computed after every 10, 25, 50, and 100 iterations respectively). This data suggests the converged accuracy is indeed not significantly impacted when employing our proposed Periodic Top K implementation.

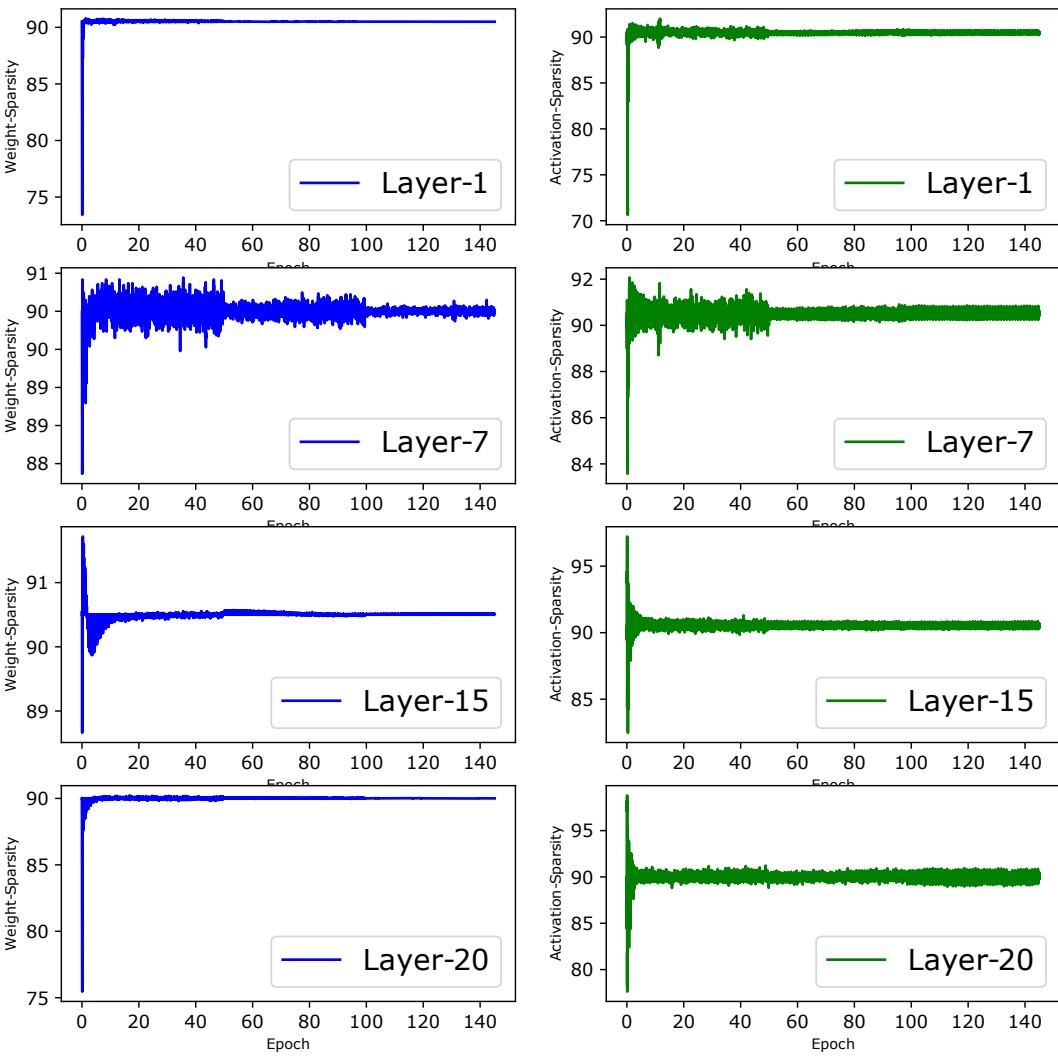

Figure 10: Sparsity Variation using Periodic Top-K Implementation. Network: ResNet-18, Dataset: CIFAR100, Top-K period: 100 iterations, Target Sparsity: 90%

Since the periodic Top-K used the same threshold during the entire period, therefore, it is crucial to confirm that periodic Top-K implementation does not adversely affect the sparsity during training. We dumped the amount of sparsity obtained in weights and activation using periodic Top-K with period 100 iteration with target sparsity of 90%. Figure10 shows the sparsity during training using periodic Top-K implementation is concentrated around our targeted sparsity, and the fluctuation decreases as training proceeds confirming our hypothesis that chosen Top-K parameter stabilizes i.e. the Top-K threshold converge to a fixed value during the latter epochs.

The Top-K periods need not to be fixed throughout the training. The Top-K period can be increased as the training proceeds because the chosen Top-K parameters are unlikely to change at the later training iterations. To demonstrate this we perform a new experiment. We train ResNet-18 on CIFAR 100 for 150 epochs (learning rate decayed at epoch 50,100) using a version of SWAT which computes Top-K periodically but with increasing Top-K period for later epochs. The Top-K schedule used during training is shown below:

| Scheduled Top-K | | Top-K | Top-1 Accuracy | |
|---|---|---|---|---|
| | | Implementation | 70% Sparsity | 90% Sparsity |
| Epoch 0-50 | 3 times per epoch | once per iteration | 76.36 | 73.63 |
| Epoch 50-100 | 1 time per epoch | Scheduled Top-K | 76.41 | 73.81 |
| Epoch 100-150 | 1 time per 5 epoch | | | |

Note: 1 epoch has 392 iterations.

**Sparsification of Batch-Normalization Layer:** The activations and weights of BN layers are not sparsified in SWAT. Empirically, we found that sparsifying weights and activations are harmful to convergence. This is because the weight (gamma) of BN layers is a scaling factor for an entire output channel, therefore, making even a single BN weight (gamma) zero makes the entire output channel zero. Similarly, dropping activations affects the mean and variance computed by BN. Empirically we found that the BN layer is extremely sensitive to changes in the per channel mean and variance. For example, when ResNet18 is trained on CIFAR 100 using SWAT with 70% sparsity and we sparsify the BN layer activations, accuracy is degraded by 4.9% compared to training with SWAT without sparsifying the BN layers. Therefore, the activations of batch-normalization layer are not sparsified.

The parameters in a BN layer constitute less than 1.01% to the total parameters in the network and the total computation in the BN layer is less than 0.8% of the total computation in one forward and backward pass. Therefore, not sparsifying batch-normalization layers only affects the activation overhead in the backward pass.

**Comparision with Lottery Ticket Hypothesis:** The Lottery Ticket Hypothesis (Frankle & Carbin, 2018) showed the difficulty of training with a sparse architecture and that sparse training is very sensitive to initial conditions. The Lottery Ticket showed if one could pick the right initial conditions for the weights, one can train with a sparse network. SWAT is interesting in that it does train a sparse network without the need for oracle information about initialization values.

We believe the crucial difference that enables SWAT to work despite the observation in the Lottery Ticket Hypothesis paper is the following. SWAT updates which weights are part of the sparse network rather than attempting to train a single unchanging sparse network. SWAT may work because it dynamically searches for the sparse architecture that will work with a given set of initial conditions.

APPENDIX C   INDEXING OVERHEAD

To estimate the indexing overhead, we need to understand the generic architecture of the sparse CNN accelerator, the sparse format in which data is stored, and how the computations are mapped to the accelerator.

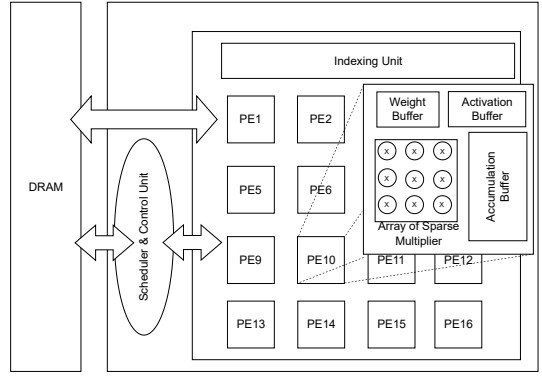

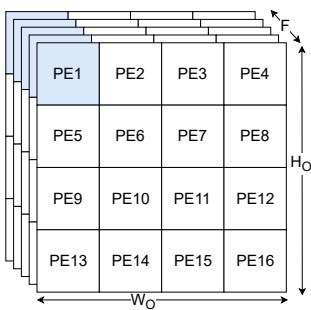

(a) Generic Architecture of Sparse CNN Accelerator

(b) Each PEs is responsible for creating a portion of the **output** (Parashar et al., 2017)

**Sparse Accelerator:** At a high-level of abstraction, all the sparse accelerators (Parashar et al., 2017; Zhang et al., 2016; Albericio et al., 2016) have a 2D array of processing units (PEs) where each processing unit has an array of multipliers and have a dedicated weight, activation, and accumulation buffer. They have an indexing unit for enabling the sparse multiplication. The computations are spatially mapped and scheduled to these processing units by a control and scheduling logic. Each of the PE generates partial products which get accumulated to compute the output values and finally stored in the DRAM.

**Mapping Computations:** Let us consider a convolutional layer, which maps the input activations $in$ ($R^{N \times C \times H_I \times W_I}$) to $out$ ($R^{N \times F \times H_O \times W_O}$). The layer computes $F$ channels of output feature maps, each of dimension $R^{H_O \times W_O}$, using $C$ channel of input feature maps of dimension $R^{H_I \times W_I}$ for each of the $N$ samples in the mini-batch. The layer has parameter $w \in R^{F \times C \times H_K \times W_K}$.

---

**Algorithm 1:** Dense Forward Pass Computation for a single input sample (Assuming Stride=1)

---

**The data:** w,in
**The result:** out
**for** $h_o$ = 1 to $H_O$ **do**
  **for** $w_o$ = 1 to $W_O$ **do**
    **for** $f$ = 1 to $F$ **do**
      **for** $c$ = 1 to $C$ **do**
        **for** $h_k$ = 1 to $H_K$ **do**
          **for** $w_k$ = 1 to $W_K$ **do**
            $c^* = c$;
            $h^* = h_o + h_k$;
            $w^* = w_o + w_k$;
            **out**$[f][h_o][w_o]+ = $ **w**$[f][c][h_k][w_k] \times $ **in**$[c^*][h^*][w^*]$);
          **end**
        **end**
      **end**
    **end**
  **end**
**end**

---

Thus, as shown in algorithm 1, each activation is reused $F \times C \times H_K \times W_K$ times, each weight is reused $N \times C \times \times H_K \times W_K$ times and the total computation is as follow:

$$Dense\ Convolution\ FLOP = F \times H_O \times W_O \times C \times H_K \times W_K \qquad (7)$$

The first three 'for' loops are independent and can be mapped independently to the PEs, whereas the inner three 'for' loop generate the partial products. The different sparse accelerators have different

ways of mapping the 'for' loops spatially over the PEs for maximizing reuse and minimizing the data transfer to and from the DRAM.

From now onwards, we are assuming the mapping of the algorithm over the PEs inspired by the SCNN architecture (Parashar et al., 2017). Each PEs is responsible for generating a chunk of output values, as shown in Figure 11b. The control unit is responsible for partitioning and transferring the corresponding input activations to each PE, whereas each PE has the entire sparse weights. Depending on the buffer size, either the entire index offset or only a block of index offset can be computed and distributed to the PEs.

**Sparse Storage Format:** In the forward pass, only weights are sparse. Each filter weights are independently stored in sparse format i.e. weights are sparsified along the dimension $R^{C \times H_K \times W_K}$. Figure 12a shows the storage of a single filter in a sparse format. The value is stored in 2 vectors; data vector and index vector. The data vectors contain only the non-zero data values, whereas the index vector contains the number of zero values before the corresponding data values.

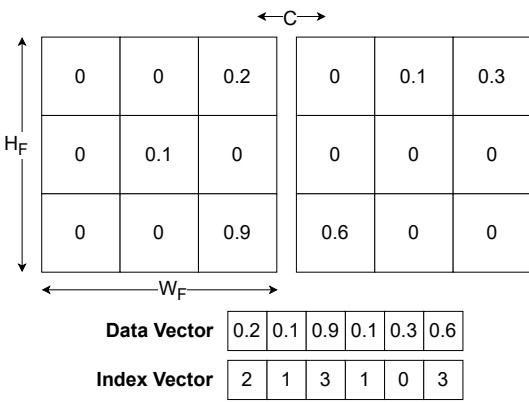

(a) Storage format for sparse weights

**Storage Overhead for Index:** Bits required for storing weights in dense format: (Assuming each weight value is stored in 32 bits)

$$Dense\ Weight\ Storage = F \times C \times H \times W \times 32 \tag{8}$$

Let the weights have sparsity $S$. The number of non-zero values present in the data vector is $(1-S) \times F \times C \times H \times W$, and the number of zero values present in the data vector is $S \times F \times C \times H \times W$. Assuming the sparsity is uniformly distributed, the number of zero values between 2 consecutive non-zero values is $\frac{S \times F \times C \times H \times W}{(1-S) \times F \times C \times H \times W} = \frac{S}{1-S}$. Therefore, the number of bits used for encoding the index vector is $max(\lceil \log_2 \frac{S}{1-S} \rceil, 1)$ bits. Taking the maximum is required when the ratio is less than one. In this case 1 bit indicates whether the next value is zero or not. The bits required for storing weights in the sparse format:

$$data\ vector = (1 - S) \times F \times C \times H \times W \times 32\ bit \tag{9}$$

$$index\ vector = (1 - S) \times F \times C \times H \times W \times max(\lceil \log_2 \frac{S}{1-S} \rceil, 1) \tag{10}$$

$$Sparse\ Weight\ Storage = (1 - S) \times F \times C \times H \times W \times [32 + max(\lceil \log_2 \frac{S}{1-S} \rceil, 1)] \tag{11}$$

Therefore, storage is decreased by a factor of $\frac{32}{(1-S) \times [32 + max(\lceil \log_2 \frac{S}{1-S} \rceil, 1)]}$. For 70% sparsity the reduction in storage space is $3.13X$.

**Index Computation Overhead:** The computation performed by each PE is shown in algorithm 1.

---

**Algorithm 1:** Sparse Forward Pass Computation for a input sample on $PE_i$(Assuming Stride=1)

---

**The data:** w, $PE_i$(in)
**The result:** $PE_i$(out)

//STEP-1: Data-Structure Initialization Phase;
$c[F][len(\textbf{index\_vector})] = 0$;
$h[F][len(\textbf{index\_vector})] = 0$;
$w[F][len(\textbf{index\_vector})] = 0$;

//STEP-2: Index Offset Calculation Phase;
**for** *f = 1 to F* **do**
    $accumulated\_index = 0$;
    **for** *ind = 1:len(**index\_vector**)* **do**
        $accumulated\_index +  = \textbf{index\_vector}[ind]$;
        $c[f][ind], h[f][ind], w[f][ind] = \text{computeIndex}(accumulated\_index)$;
    **end**
**end**

//STEP-3: Computations Phase;
**for** $h_o$ *= 1 to* $PE_i(H_O)$ **do**
    **for** $w_o$ *= 1 to* $PE_i(W_O)$ **do**
        **for** *f = 1 to F* **do**
            **for** *ind = 1:len(**index\_vector**)* **do**
                $c^* = c[f][ind]$;
                $h^* = h_o + h[f][ind]$;
                $w^* = w_o + w[f][ind]$;
                $\textbf{out}[f][h_o][w_o] + = \textbf{data\_vector}[ind] \times PE_i(\textbf{in})[c^*][h^*][w^*]$;
            **end**
        **end**
    **end**
**end**

---

The indexing involved during the Computation Phase is just a base+offset addition. The same kind of index computation is present in dense convolution(algorithm 1). Therefore, the only additional indexing computation is during the Indexing Offset Calculation Phase. Let the weights have sparsity $S$.

$$Index\ Computation\ OP = \alpha \times S \times F \times C \times H_K \times W_K \tag{12}$$

where $\alpha$ is the overhead of the computeIndex function. computeIndex function is accumulating the current index and calculating the 3 offset ($c[f][ind], h[f][ind], w[f][ind]$). Therefore, the value of $\alpha$ will be approximately around 4.

$$Sparse\ Computation\ FLOP = S \times F \times H_O \times W_O \times C \times H_K \times W_K \tag{13}$$

Therefore,

$$Total\ Operation\ for\ Sparse\ Convolution = S \times F \times C \times H_K \times W_K \times (\alpha + H_O \times W_O) \tag{14}$$

Since the $H_O \times W_O >> \alpha$, the indexing overhead is minimal compared to the benefit obtained by performing sparse convolution.

**Chip Area Consideration:** Considering the chip area (#$Transistors$ available) is very critical when designing a custom accelerator since it decides the number of the computational units that can be fabricated on a given area. Generally, the number of transistors needed for a dense computation unit is less than that of the sparse computation unit. This is because of the extra chip area required for the indexing, arbitration, and controlling logic. Based on the area value mentioned in the SCNN paper(Parashar et al., 2017), the scaling factor for adjusting the performance of the sparse CNN

accelerator is $0.75$ i.e., On a given area, we can have almost $1.33\times$ more dense computation units compared to sparse computation units.

**Backward Pass:** The computation in the backward pass is the deconvolution operation. Since deconvolution operation is very similar to the convolution operation therefore the overhead of index computation in the backward pass would be of the same order compared to the computation in the forward pass.

