# OpenReview forum: "Sparse Weight Activation Training"
_ICLR.cc/2020/Conference — Reject_

### Official Review · AnonReviewer3 · 2019-10-12
**Official Blind Review #3**

**Rating:** 6

**Review:**

This paper studies training neural networks with sparse weights and sparse activations (SWAT training). By using sparse weights in forward passes as well as sparse weights and activations in backward passes, SWAT can reduce the computation overhead and also reduce the training memory footprint. The primary contributions of the paper are in three folds: 1) The authors empirically compare the impact of (activation) gradient sparsity and weight + activation sparsity on the model performance---the comparison shows that the weight + activation sparsity has less influence on the model accuracy; 2) Across different models on CIFAR and ImageNet dataset, SWAP can reduce the training flops by 50% to 80% while using roughly 2 to 3x less training memory footprint saving (weight + activation); 3) The authors empirically study on why training using top-K based sparsification can attain strong model accuracy---the magnitude-based top-K approach can roughly preserve the directions of the vectors.

I think the claimed contributions are well-validated in general. The design decisions of the approach are well supported by empirical observations and the components of the approach (different top-K methods) are studied properly. Additionally, I like the authors' synthetic-data studies to shed light on why top-K based sparsity can work well. Given the above reason, I give week accept and I am willing to raise the score if the following questions / concerns can be resolved in the rebuttal / future draft:

1. In results such as in figure 4, we observe that using intermediate levels of sparsity can actually demonstrate better generalization performance than the dense baseline training approach. I was wondering if this is because the default hyperparameter produces better training loss in sparse training than in dense training, and consequently the sparse training test performance is also improved over dense training. Without showing this, it is not fully convincing that intermediate sparsity helps prevent overfitting and generalizes better (as the authors discussed in the text).

2. For "Impact on Convergence" in section 3.2, it is not clear to me what the authors are using as a metric for the degree of convergence. Thus I can not evaluate the claims here.

3. For "Efficient Top K implementation" in section 3.2, the authors suggest  only computing the K-th largest elements periodically to further improve efficiency. However the empirical evidence of whether this approach will significantly degrade the model performance at the end of training is not provided.

4. For the GFLOPS comparison in Figure 7, could the authors elaborate what operations are included into the count? As the sparse operations requires additional indexing operations for computation, I was wondering whether the GFLOPS can realistically reflect the real latency / energy efficiency of the SWAT approach.

5. How the memory access count calculated at the end of page 7? Is it counting the number of float point values (activations, activation gradients, weights) that needs to be fetched for forward and backward pass?

6. At the first paragraph in page 8 (last paragraph above section 4), do the authors imply that the activations of BN layers is not sparsified? Could the authors provide a bit more evidence on how (any why) sparsification of BN activation impacts the model performance.



**Experience Assessment:**

I have read many papers in this area.

**Review Assessment: Checking Correctness Of Derivations And Theory:**

I carefully checked the derivations and theory.

**Review Assessment: Checking Correctness Of Experiments:**

I carefully checked the experiments.

**Review Assessment: Thoroughness In Paper Reading:**

I read the paper at least twice and used my best judgement in assessing the paper.

---

> ### Author Response · Authors · 2019-11-09
> **Response to Reviewer 3**
>
>
> 4. Figure 7 includes all the operations for the convolutional layer, batch-normalization layer, linear layer and the Top-K operation assuming Top-K is implemented using BFRT+thresholding operation.  The GFLOPs calculation doesn’t consider how these operations would be implemented in sparse format.
>
> We believe SWAT would be well suited for emerging sparse accelerator hardware designs that contain dedicated hardware (e.g., for indexing). There have been many recent publications describing such sparse accelerators [1-3] and they show significant performance and energy improvement even in the presence of irregular sparsity.  We have left evaluation of runtime of SWAT on such accelerators as future work.
>
> [1] SCNN: An Accelerator for Compressed-sparse Convolutional Neural Networks
> [2] Cambricon-X: An accelerator for sparse neural networks
> [3] Cnvlutin: Ineffectual-neuron-free deep neural network computing
>
> 5.  Figure 8(a) shows the memory access reduction for weights only and Figure 8(b) shows the reduction for activations only, both only for the backward pass. Depending on the batch size, weights and activations represent 50-70% of total memory traffic in the backward pass.  SWAT does not reduce the overhead of activation gradients, but these are immediately consumed (when evaluating the prior layer during backprop) and could therefore potentially (depending on batch-size) even be stored in on-chip memory.
>
> 6. Yes, activations and weights of BN layers are indeed not sparsified in SWAT. Empirically, we found that sparsifying weights and activations are harmful to convergence. This is because the weight (gamma) of BN layers is a scaling factor for an entire output channel, therefore, making even a single BN weight (gamma) zero makes the entire output channel zero. Similarly, dropping activations affects the mean and variance computed by BN. Empirically we found that the BN layer is extremely sensitive to changes in the per channel mean and variance. For example, when ResNet18 is trained on CIFAR 100 using SWAT with 70% sparsity and we sparsify the BN layer activations, accuracy is degraded by 4.9% compared to training with SWAT without sparsifying the BN layers. Therefore, the activations of batch-normalization layer are not sparsified.
>
> The parameters in a BN layer constitute less than 1.01% to the total parameters in the network and the total computation in the BN layer is less than 0.8% of the total computation in one forward and backward pass. Therefore, not sparsifying batch-normalization layers only affects the activation overhead in the backward pass. Currently we are working on reducing the activation overhead for the batch-normalization layer.

---

> > ### Comment · AnonReviewer3 · 2019-11-14
> > **Thanks for addressing my comments**
> >
> > Dear authors,
> >
> > Thanks for the efforts in resolving my comments.
> >
> > I think the discussions in the rebuttal will be very helpful for readers to better understand the contribution. And I would suggest to add the discussions to the appendix.
> >
> > I will keep the current rating and I will raise the score to 7 if the following results can be presented:
> >
> > The authors mentioned the indexing overhead can be small in the emerging architectures for sparse accelerators. Could the authors uses some of the efficiency reports in the referred papers to give us a more grounded estimates on efficiency. My only concern here is that FLOPS is not the most trustworthy metric in many scenarios [1], and the domain readers might be more convinced a better evaluation metric like (estimated) latency can be reported.
> >
> > [1] ProxylessNAS: Direct Neural Architecture Search on Target Task and Hardware. Cai et al.

---

> > > ### Author Response · Authors · 2019-11-15
> > > **Thank you for your response**
> > >
> > > We have added most of the rebuttal discussions either in the appendix or clarified in the paper.  We have added a detailed performance estimation on the sparse accelerator in Appendix C.
> > >
> > > Summary:
> > > Storage Overhead: Under the assumption of uniformly distributed sparsity, the storage overhead of index array grows logarithmically with the sparsity. The exact growth factor is Log2(S/(1-S)) where S is the sparsity.
> > >
> > > Computation Overhead:  The ratio of index computation overhead to the sparse computation is inversely proportional to the output channel size. The proportionality constant is a property of the accelerator and is independent of the weights and input activations size. Therefore, the overhead is small compared to the benefit obtained by sparse computations. The full derivation is in the Appendix C.

---

> ### Author Response · Authors · 2019-11-09
> **Response to Reviewer 3**
>
> Thank you for your feedback and questions. We provide answers below.
>
> 1. The main point of our paper is to reduce computation required to train.  The effect we observed that accuracy improved was only seen for small datasets (like CIFAR 10).  The results in our submission are based upon three runs.  We have run more experiments and believe the effect seen actually may simply have arisen due to random variation (due to different random seeds).  As the improvements that were shown were small and not the main point of our paper we will remove the claim about improved accuracy from the revised paper and include revised figures averaging over more runs.  That said, to answer the question the training accuracy for these cases was uniformly close to 100% for dense and SWAT up to 40% sparsity.
>
> 2. The metric we are using for measuring “degree of convergence” is the number of epochs it takes to reach the saturation accuracy.  So our observations that the rate of convergence is not impacted follow from our observation that the change in accuracy from one epoch to the next for validation accuracy saturates around the same epoch for both SWAT and dense training. As shown in Figure 5, when the learning rate is 0.1 (i.e. between epoch 0 and 30) the SWAT algorithm reaches the saturation accuracy around the 15th epoch approximately the same epoch when the baseline algorithm also reaches saturation. Similarly, when the learning rate is 0.01 (i.e. between epoch 0-40th)  both SWAT and the baseline saturate at epoch 35th.
>
> Second, we want to clarify that we do not use any early stopping criteria but rather run a fixed number of epochs.
>
>
> 3. We have empirically confirm this approach works as follows.  We define “Top-K period” as the number of iterations between computing the threshold for Top-K.  The table below shows top1 validation accuracy from single runs for CIFAR 100 on ResNet-18 with different Top-K periods (i.e., Top-K is computed after every 10, 25, 50 and 100 iterations respectively). This data suggests the converged accuracy is indeed not impacted significantly (if at all) when employing our proposed efficient Top K implementation.
>
> 			Sparsity 70%		Sparsity 90%
>    1 iteration:	     	     76.41		     73.81
>    10  iteration:   	     76.59		     73.64
>    20  iteration:            76.03		     73.45
>    50  iteration:            76.06		     74.09
>    100  iteration:	     76.52		     73.29

---

### Official Review · AnonReviewer1 · 2019-10-24
**Official Blind Review #1**

**Rating:** 3

**Review:**

This paper proposes SWAT as a training algorithm for sparse networks on different architectures. The paper claims being able to reach a level of sparsity with no drop in accuracy.  The goal is to minimize the computations during training time. To this end, SWAT sets to zero the vectors where necessary. Different from other approaches, SWAT uses sparse computation in the forward and backward passes. The intuition behind is that eliminating small components does not have an impact on the training process but can be used to minimize the computation required.

Some Comments:

- The paper is a bit on the empirical side with a decent number of experiments to demonstrate the effectiveness of the proposal. I am on the border between accepting and rejecting.


- The top-K implementation is interesting. Page 7 suggests the top-K do not change during training which is reasonable as the update is limited to those components. Would it be possible to avoid completely that compute and quickly select K early in the training process? I would find that an interesting future direction.

- In the experimental section, I missed actual numbers. At the moment, if I understand correctly, the paper is based on theoretical compute savings. How feasible is this considering the sparsity of the operation (assuming unstructured sparsity)?

- In the case of structured sparsity, how this differs from the early pruning process of regularization based pruning algorithms? For instance, in the first reference (compression-aware training), the authors claim the model can be compressed in the early training. If that is the case, how different is SWAT from those type of methods? In those related works, the accuracy does not drop. Implementation wise, those algorithms do make the backward pass also sparse (setting to 0 the gradients).

- At the moment, the algorithm is using a magnitude-based sorting. Would it be possible to have other sorting approaches?


Minor things:

- for clarity, I would summarize the algorithm in section 2.2 rather than in the appendix.

- I am surprised by the imagenet training setting. Why only training for 50epochs? The standard training process is 90epochs changing the learning rate in the 30th and 60th.

- I guess the S% sparsity contribution can be improved (rephrased). If the training algorithm sets to zero N parameters seems to me obvious that there will be no drop in accuracy compared to that training process. What is the drop in accuracy referring to?

- check the references. While the list is quite comprehensive, some of them are not referred in the text. Please, add comments where appropriate.

**Experience Assessment:**

I have published in this field for several years.

**Review Assessment: Checking Correctness Of Derivations And Theory:**

I assessed the sensibility of the derivations and theory.

**Review Assessment: Checking Correctness Of Experiments:**

I assessed the sensibility of the experiments.

**Review Assessment: Thoroughness In Paper Reading:**

I read the paper at least twice and used my best judgement in assessing the paper.

---

> ### Author Response · Authors · 2019-11-09
> **Response to Reviewer 1**
>
> Thank you for your feedback and questions. We provide answers below.
>
> Some Comments
> 1  First, we want to clarify that Top-K weights and activations are used for computing gradient but dense gradients are used for updating the weights.  Specifically, the convolution between the sparse input activation and dense output activation gradient  will create a dense weight gradient.  Thus, updates are not limited to only the Top-K components. Even parameters that have been dropped during the forward and backward pass get updated i.e. the entire weight gradient is used to update the parameters and there is no masking of the weight gradient.
>
> Thus, SWAT doesn’t perform hard elimination of the parameter and it allows the algorithm to capture the dynamic sparsity present in the model since a parameter eliminated at some early iteration may come back in the later training iteration because of the updates.
>
> The SWAT algorithm is trying to capture the dynamic sparsity in the model, therefore the Top-K operation should be performed periodically but the period can be increased as the training proceeds because the chosen Top-K parameters are unlikely to change at the later training iterations. To demonstrate this we perform a new experiment.  We train ResNet-18 on CIFAR 100 for 150 epochs (learning rate decayed at epoch 50,100) using a version of SWAT which computes Top-K periodically but with increasing Top-K period  for later epochs. The Top-K schedule used during training is shown  below:
>
> Top-K period
> Epoch       0 to 50:    three times per epoch
> Epoch   50 to 100:     once per epoch
> Epoch 100 to 150:  once per 5 epochs
>
> The below table shows Top-1 Accuracy:
> Top-K period			               Sparsity 70% 	 	Sparsity 90%
> Above Top-K schedule                  76.36		     73.63
> 1 iteration:	     	                          76.41		     73.81
>  Note: 1 epoch has 392 iterations
>
> 2- Yes, the paper reports only the theoretical compute and memory bandwidth reduction. The exact performance and energy benefit would be proportional to the reported savings but will depend on the underlying hardware. Note: achieving practical speed-up on GPUs would be difficult because of the unstructured sparsity but the speed-up could be obtained on CPUs and sparse accelerators. There are many works which have proposed sparse accelerators for exploiting unstructured sparsity such as [1-3].
>
> [1] SCNN: An Accelerator for Compressed-sparse Convolutional Neural Networks
> [2] Cambricon-X: An accelerator for sparse neural networks
> [3] Cnvlutin: Ineffectual-neuron-free deep neural network computing
>
> 3- All of the regularization based pruning work focused on reducing the network weights by using regularizer during training for promoting weight sparsity. Sparse weights will only accelerate one part of backward pass i.e. input activation gradient computation will be fast but not the weight gradient computation. The weight gradient computation will still involve convolution between dense activations and dense output activation gradients. The computation in both parts of the backward pass is roughly of the same order as shown below
>
>  			        WeightGradientComputation  InputActivationGradientComputation
> 	ResNet-18		        3.6   GFLOP			5.5  GFLOP
> 	VGG-16		                30.7 GFLOP			30.7GFLOP
> 	DenseNet-121		5.7   GFLOP			6.4  GFLOP
> Therefore, speedup for early regularization based pruning would be limited by the weight gradient computation. Moreover, such works generally use more expensive optimization methods such as proximal gradient descent compared to standard gradient descent.
>
> 4- Yes, for example, instead of sorting individual values, channel-based sorting method could be used. The importance of channel could be measured or learned during the training itself. One other sorting approach would be to consider the parameter as well as its gradient during sorting. These are interesting future directions and should be investigated but it is beyond the scope of our current work.
>
> Second, these methods do not lose accuracy since their compression ratio is small, for example only 27% compression in compression-aware training for ImageNet dataset; whereas, SWAT achieves much higher compression rate such as 50% with only 0.26-1.1% loss in accuracy for ImageNet. Note: SWAT training does not employ any fine-tuning so users may perform fine-tuning later to further reduce the gap.

---

> ### Author Response · Authors · 2019-11-09
> **Response to Reviewer 1 (Minor Things)**
>
> Thank you for your feedback and questions. We provide answers below.
> Minor Comments
> 1- We will summarize the algorithm in section 2.2 in the final version.
>
> 2- We wanted to decrease the training time for running many experiments. Experimentally, we found that the network learns the most in the first learning regime (epoch 0 to 30) and therefore the network should be trained for the same number of epochs for the first learning regime. In the second (epoch 30 to 60) and the third learning regime (epoch 60 to 90), we realized that the accuracy attained after 10 epochs is closer to the saturation accuracy attained in that learning regime. Therefore, we decrease the duration of the second and third learning regime to only 10 epochs instead of 30 epochs each, thereby reducing total epoch by 40 which reduces the training time by 44%.  If needed we will put the result for training for 90 epochs for some architecture.
>
> 3- Could you please elaborate and let us know which section of the paper are you referring to?
>
> 4- We will cite and add necessary comments for any of the missing references.

---

> > ### Author Response · Authors · 2019-11-15
> > **Thank you for your review**
> >
> > We have resolved all of your mentioned issues and have added most of the rebuttal discussions either in the appendix or clarified in the paper. As per asked by you and reviewer 3, we have added a detailed performance estimation on the sparse accelerator in Appendix C.

---

### Official Review · AnonReviewer4 · 2019-11-13
**Official Blind Review #4**

**Rating:** 3

**Review:**

I am the emergency reviewer. Sorry for the late.

This paper studies a very interesting topic: eliminating small magnitude components of weight and activation vector instead of eliminating small magnitude components of gradients. A clear interpretation and definition towards the forward and backward propagation is presented. The difference between meProp versus SWAT is also plain. Based on some experiment results shown in Figure2, authors announced that accuracy is extremely sensitive to sparsification of output gradients. Thus algorithms SWAT and SAW are proposed to prune the model, which are respectively training with sparse weights and activations, and SWAT only sparsifies the backward pass. Top-K selection is implemented to select which components are set to zero during sparsification.

Strengths:
1. The writing logic ascends step by step.
2. Authors showed the harmfulness of the sparsity of gradients by experiment results. Also the comparison between the sparsity of weights and activations are meaningful.
3. Sufficient experiments are done to generalize SWAT to different models, and the results are fascinating on ImageNet.

Weaknesses:
It's a borderline paper.
1. lack of novelty. The paper has shown a lot experiment results on basic models, but the raising of Top-K algorithm is not novel. Why it is Top-K but not other metrics for selecting zero components? In this view, the paper is likely to be a project summary.
2. Less comparison to other basic pruning models. More experiments should be done to compare SWAT with other sota pruning models. Then the results will be convincing.


**Experience Assessment:**

I have published one or two papers in this area.

**Review Assessment: Checking Correctness Of Derivations And Theory:**

I assessed the sensibility of the derivations and theory.

**Review Assessment: Checking Correctness Of Experiments:**

I assessed the sensibility of the experiments.

**Review Assessment: Thoroughness In Paper Reading:**

I read the paper at least twice and used my best judgement in assessing the paper.

---

> ### Author Response · Authors · 2019-11-13
> **Response to Reviewer 4**
>
> Thank you for your feedback and questions.
>
> First, we would like to clarify that the objective of the paper is not to prune the model but rather to train the network faster (with fewer computations and less memory bandwidth). We just happen to do that by sparsifying computation.  SWAT does indeed have the side effect of pruning the network, but it does not do so nearly as well as papers that set out with pruning as their primary objective which was never the goal of our work.  To the best of our knowledge, the large body of literature on pruning increases training time since those works either employ an expensive 3-stage pruning pipeline or introduce additional computations for computing parameter saliency. There is a large body of work that tries to accelerate training by reducing the number of iterations to reach convergence (e.g., Batch Normalization, ADAGRAD, ADAM, RMSPROP). Our work differs in that it focuses on how to reduce the computation per iteration. Prior work we are aware of that tackles the problem of accelerating training by reducing computation per iteration are meProp [a] and DSG [b] which we compared against in the paper.
>
> Second, SWAT sparsifies the entire training process i.e. sparsifies the forward as well as the backward pass.
>
> Novelty:
> Our novelty is showing that the network can be trained with high sparsity without loss in accuracy and the algorithm generalizes well even to complex architectures such as  deeper and wider networks on large datasets.
>
> While not stated explicitly in the current draft, we believe the following is an important and novel aspect of our submission. At ICLR 2019 the Lottery Ticket Hypothesis [c] (best paper winner) showed the difficulty of training with a sparse architecture and that sparse training is very sensitive to initial conditions.  The Lottery Ticket showed if one could pick the right initial conditions for the weights, one can train with a sparse network.  SWAT is interesting in that it does train a sparse network without the need for oracle information about initialization values.
>
> We believe the crucial difference that enables SWAT to work despite the observation in the Lottery Ticket Hypothesis paper is the following.  SWAT updates which weights are part of the sparse network rather than attempting to train a single unchanging sparse network.  SWAT may work because it dynamically searches for the sparse architecture that will work with a given set of initial conditions.
>
> Why it is Top-K function
> In the appendix we have shown that the Top-K function is the sparsifying function which causes minimum deviation in the cosine distance between original vector and the sparsified instance of that vector. Therefore we have used the Top-K function for sparsification.  There are works such as [d] [e] which has shown that cosine similarity is a useful metric for measuring convergence.
>
> Comparison to other state of the art pruning models:
> Again, our objective is NOT to prune the model (at any training cost) but rather accelerate the training of a given network. Pruning was just an interesting by-product of our algorithm.  We believe if one is concerned with model compression, one could take the resulting network achieved by using SWAT (more quickly trained) and then apply one of the many pruning algorithms to it.  In other words, the works are orthogonal.
>
> [a]  meprop: Sparsified back propagation for accelerated deep learning with reduced overfitting.
> [b] Dynamic Sparse Graph for Efficient Deep Learning.
> [c] The Lottery Ticket Hypothesis: Finding Sparse, Trainable Neural Networks.
> [d] Scalable Methods for 8-bit Training of Neural Networks.
> [e] The High-Dimensional Geometry of Binary Neural Networks.

---

> > ### Author Response · Authors · 2019-11-15
> > **Thank you for your review**
> >
> > We have added the comparison with the lottery ticket hypothesis in the appendix.  Also, we have added a detailed performance estimation on the sparse accelerator in Appendix C.

---

### Decision · Program_Chairs · 2019-12-19

**Decision:**

Reject

**Comment:**

The paper is proposed a rejection based on majority reviews.